# Kupczynski's Contextual Locally Causal Probabilistic Models Are Constrained by Bell's Theorem

Richard D. Gill [1,*] and Justo Pastor Lambare [2]

1   Mathematical Institute, P.O Box 9512, 2300 RA Leiden, The Netherlands
2   Facultad de Ciencias Exactas y Naturales, Universidad Nacional de Asunción, Ruta Mcal. J. F. Estigarribia, Km 11 Campus de la UNA, San Lorenzo 111241, Paraguay; jupalam@gmail.com
*   Correspondence: gill@math.leidenuniv.nl

**Abstract:** In a sequence of papers, Marian Kupczynski has argued that Bell's theorem can be circumvented if one takes correct account of contextual setting-dependent parameters describing measuring instruments. We show that this is not true. Despite first appearances, Kupczynksi's concept of a contextual locally causal probabilistic model is mathematically a special case of a Bell local hidden variables model. Thus, even if one takes account of contextuality in the way he suggests, the Bell–CHSH inequality can still be derived. Violation thereof by quantum mechanics cannot be easily explained away: quantum mechanics and local realism (including Kupczynski's claimed enlargement of the concept) are not compatible with one another. Further inspection shows that Kupczynski is actually falling back on the detection loophole. Since 2015, numerous loophole-free experiments have been performed, in which the Bell–CHSH inequality is violated, so, despite any other possible imperfections of such experiments, Kupczynski's escape route for local realism is not available.

**Keywords:** Bell's theorem; local realism; quantum entanglement; contextuality; Bell–CHSH inequality

## 1. Introduction

Marian Kupczynski ("MK") is the author of a thought-provoking paper published (2021) in the journal *Entropy* [1] and entitled "Contextuality-by-default description of Bell tests: contextuality as the rule and not as an exception". The work, as are some other papers by the same author, is built around a mathematical claim by MK which we argue is incorrect. Misunderstandings of Bell's theorem came up almost immediately after Bell's first proof of his theorem was published. They are still widespread, despite Bell's published first replies [2] to his critics, and his later more detailed published accounts of his theorem, such as that in [3]. Already, we published [4] a short "Comment" on the one year earlier paper [5] published by MK, in which MK made the same claims, and MK rapidly published a response [6]. In this paper, we again present (for ease of the reader) our counterexample to the claim that Bell's theorem does not hold for what MK calls a *Contextual Locally Causal Probabilistic Model* of an EPRB experiment. We, moreover, add an alternative construction expressed in the framework of modern statistical causality theory, based on what one can also call a classical locally causal probabilistic model, described by a DAG (Directed Acyclic Graph), a graph with nodes representing variables, some of which are observed and some only conjectured to exist. Directed links between nodes stand for direct probabilistic causal dependences. MK's framework, expressed as a DAG, has already been used in the literature on Bell inequalities and Bell's theorem. One particular example is a well-known paper by Spekkens and Wood (2015) [7].

In the abstract of [1], we read "[the] system of random variables 'measured' in Bell tests is inconsistently connected and it should be analyzed using a Contextuality-by-Default approach, what is done for the first time in this paper". He had actually done that in earlier papers, but probably the order in which those papers was written differs from the order in which they were published. We take as our starting point the very extensive abstract of

Kupczynski's paper [5] published one year before [1]. For the reader's convenience, we reproduce it here, separated into three sections. First comes an introduction, the reference to spreadsheets therein is perhaps inspired by a paper by one of us, [8].

> *Bell–CHSH inequalities [named for the Clauser, Horne Shimony, Holt generalization of Bell's original inequality] are trivial algebraic properties satisfied by each line of an $N \times 4$ spreadsheet containing $\pm 1$ entries, and, thus, it is surprising that their violation in some experiments allows us to speculate about the existence of non-local influences in nature and casts doubt on the existence of the objective external physical reality. Such speculations are rooted in incorrect interpretations of quantum mechanics and in a failure of local realistic hidden variable models to reproduce quantum predictions for spin polarization correlation experiments (SPCEs). In these models, one uses a counterfactual joint probability distribution of only pairwise measurable random variables $(A, A', B, B')$ to prove Bell–CHSH inequalities.*

Next, MK explained what was for him the heart of the matter:

> *In SPCE, Alice and Bob, using 4 incompatible pairs of experimental settings, estimate imperfect correlations between clicks registered by their detectors. Clicks announce the detection of photons and are coded by $\pm 1$. Expectations of corresponding random variables – $E(AB)$, $E(AB')$, $E(A'B)$, and $E(A'B')$ – are estimated and compared with quantum predictions. These estimates significantly violate CHSH inequalities. Since variables $(A, A')$ and $(B, B')$ cannot be measured jointly, neither $N \times 4$ spreadsheets nor a joint probability distribution of $(A, A', B, B')$ exist, thus Bell–CHSH inequalities may not be derived. Nevertheless, imperfect correlations between clicks in SPCE may be explained in a locally causal way, if contextual setting-dependent parameters describing measuring instruments are correctly included in the description.*

Finally, he presented his own metaphysical conclusions regarding quantum interpretations and the foundations of quantum mechanics:

> *The violation of Bell–CHSH inequalities may not therefore justify the existence of a spooky action at the distance, super-determinism, or speculations that an electron can be both here and a meter away at the same time. In this paper we review and rephrase several arguments proving that such conclusions are unfounded. Entangled photon pairs cannot be described as pairs of socks nor as pairs of fair dice producing in each trial perfectly correlated outcomes. Thus, the violation of inequalities confirms only that the measurement outcomes and 'the fate of photons' are not predetermined before the experiment is done. It does not allow for doubt regarding the objective existence of atoms, electrons, and other invisible elementary particles which are the building blocks of the visible world around us.*

The Bell–CHSH inequalities certainly can be seen as trivial algebraic properties of some very elementary mathematical structures. However, what MK calls *speculations* are actually *arguments* which depart from physical assumptions. MK's general objections were already answered by John Bell himself in Bell (1975) [2], and the assumptions behind the Bell–CHSH inequalities have been clearly stated by Bell (1981) [3]. We come back to these matters later.

Our focus is on what MK sees as the content of Bell's theorem, and which he summarizes as "the failure of local realistic hidden variable models to reproduce quantum predictions for spin polarization correlation experiments (SPCE)". MK's concept of "local realistic hidden variables models" is basically Bell's original specification in his landmark 1964 paper (itself inspired by EPR). In that first paper, Bell's use of perfect anti-correlation was to argue for realism. Then, thanks to perfect anti-correlation at equal settings, the principle of locality was used to argue that, in the idealized EPR–B experiment, all correlation and all randomness must derive from hidden variables "originally located at the source". However, this was a physical interpretation of the mathematical model which Bell presented. It is not the only physical interpretation. MK only *appears* to depart from this by introducing further hidden variables "located in the measurement devices" and with probability distributions which may depend on the local setting. However, allowing

contextual setting-dependent parameters in this way does not actually extend the sets of probability distributions of possible measurements allowed in the standard, and only apparently restrictive, "local realistic hidden variable models". His only apparently broader framework cannot, therefore, reproduce the standard quantum predictions for "spin polarization correlation experiments", since any sets of correlations allowed within his extended model necessarily satisfy the Bell–CHSH inequalities.

We show that MK's framework does enable the construction of joint probability distributions of measurement outcomes under different possible settings. It enables exactly the same collection of joint probability distributions to be realized in his mathematical framework as those enabled by Bell's original mathematical framework. The possibility of extracting probability distributions of outcomes of feasible experiments from probability distributions of joint outcomes of infeasible experiments is called "counterfactual definiteness" by some. The phrase "counterfactual variables" then leads to the objection that the concept is unphysical. We know from quantum mechanics that one cannot measure spin in different directions simultaneously, and, hence, within quantum mechanics it makes no physical sense whatsoever to introduce such variables. We comment on this debate towards the end of this paper. For now, we just remark that it is irrelevant to our primary claim: the Bell–CHSH inequalities do hold in Kupczynski's framework. It, therefore, cannot reproduce the standard predictions for the EPR–B model. It cannot explain how Bell inequalities are violated in real experiments.

John Bell once remarked "proofs of what is impossible often demonstrate little more than their authors' own lack of imagination". It is essential to distinguish the mathematical issue of existence of joint probability distributions with given lower dimensional marginal distributions from the metaphysical issue of the physical existence of "counterfactual" variables. What is unimaginable from the point of view of quantum mechanics was imaginable from the point of view of EPR.

Our paper also contains a discussion of further issues in MK's paper. On some matters, we find ourselves able to agree with MK, for the completely opposite reason! Specifically, because of Bell's theorem, we agree with MK that violation of Bell inequalities is a strong indication that "entangled photon pairs cannot be described as pairs of socks nor as pairs of fair dice" and also that "the measurement outcomes are not predetermined".

Our findings have consequences for some earlier papers by MK [9–11] presenting the same ideas in more or less detail. We chose to focus on [1,5] as being the most recent and comprehensive exposition of MK's results and ideas. Similar ideas have been developed by several other authors, such as, in particular, Karl Hess, Hans de Raedt, Andrei Khrennikov, and Theo Nieuwenhuizen. Their works are all cited in [1,5] and, to keep this paper brief, we do not present and critique their contributions (in as much as they follow an approach similar to MK's) here.

In a yet more recent paper [12], Kupczynski takes a new tack. The joint probability distribution of local setting-dependent hidden variables is allowed to depend on *both* settings in a completely arbitrary way. This mathematical model allows completely arbitrary joint probability distributions of the outcomes given the settings, hence it can reproduce any four correlations whatsoever in a Bell experiment.

However, this model is nonlocal. This becomes evident using a more convenient notation, changing $p_{ij}(\lambda_i, \lambda_j)$ to $p(i, j, \lambda_i, \lambda_j)$, MK's condition $p_{ij}(\lambda_i, \lambda_j) \neq p_i(\lambda_i)p_j(\lambda_j)$ becomes

$$p(i, j, \lambda_i, \lambda_j) = p(i, \lambda_i \mid j, \lambda_j)p(j, \lambda_j) \neq p(i, \lambda_i)p(j, \lambda_j) \tag{1}$$

So, $p(i, \lambda_i \mid j, \lambda_j) \neq p(i, \lambda_i)$ which simply means that we are transferring the violation of local causality from the measurement outcomes to the instrument setting choices.

In Appendix A to this paper we discuss a section of [1] where MK reproduces Bell's own treatment of instrument hidden variables (showing that they were also covered by his model). MK recognises that Bell's formulae reproduce the same correlations as MK's "contextual model". His objection to Bell's analysis is that the mathematical derivation, according to MK, does not correspond to any actual experimental procedure. We fail to

see how this can be a reason to ignore the purely. mathematical consequences of the initial assumptions.

## 2. A Fallacious Argument

Before going into what Kupczynski calls his "contextual model" we highlight an incorrect inference already appearing in the abstract of [5], in which he suggests that Bell inequalities cannot be proven:

*Since variables $(A, A')$ and $(B, B')$ cannot be measured jointly, neither $N \times 4$ spreadsheets nor a joint probability distribution of $(A, A', B, B')$ exist, ......*

Indeed, according to quantum mechanics, noncommuting observables cannot be measured simultaneously. However, this fact is irrelevant. The fact that there is no logical impediment to the eventual existence of a joint probability distribution, even though measurements $(A, A')$ and $(B, B')$ are physically impossible, can be proved with a counterexample. Use of Fine's theorem [13] makes this job easy. In [14], a counterexample is constructed where we have four incompatible random experiments à la Bell satisfying the CHSH inequalities. Hence, according to Fine's theorem, they admit a joint distribution with appropriate marginals reproducing the "actual" experimental probability distributions.

For the sake of completeness, we reproduce the counterexample here. Suppose Alice and Bob are experimenters living in two separate cities, say Paris and Seoul. A third experimenter, named Charlie and living in Kabul, acts as the source. The hidden variable is obtained when Charlie throws a die in Kabul obtaining the value $\lambda \in \{1, 2, 3, 4, 5, 6\}$. Each time Charlie obtains a value of $\lambda$, he transmits it through a classical channel to Alice and Bob. In their respective laboratories, after receiving a value of $\lambda$, each experimenter tosses a fair coin and evaluates the functions $A(a, \lambda)$ and $B(b, \lambda)$

$$A(a, \lambda) = a^{\lambda} \tag{2}$$
$$B(b, \lambda) = b^{\lambda+1} \tag{3}$$

where $a, b \in \{+1, -1\}$ with *Heads* $\equiv +1$ and *Tails* $\equiv -1$. The two possible values of $a$ and $b$ are incompatible; a coin gives either *Heads* or *Tails* but not both, just as in the CHSH spin experiment where each party can measure only one direction but not both. We have

$$E(A_1 B_1) = \sum_{i=1}^{6} p(\lambda_i) 1^{\lambda_i} 1^{\lambda_i+1} = \frac{1}{6} * 6 = 1 \tag{4}$$

$$E(A_{-1} B_1) = \sum_{i=1}^{6} p(\lambda_i) (-1)^{\lambda_i} 1^{\lambda_i+1} = \frac{1}{6} * 0 = 0 \tag{5}$$

$$E(A_1 B_{-1}) = \sum_{i=1}^{6} p(\lambda_i) 1^{\lambda_i} (-1)^{\lambda_i+1} = \frac{1}{6} * 0 = 0 \tag{6}$$

$$E(A_{-1} B_{-1}) = \sum_{i=1}^{6} p(\lambda_i) (-1)^{\lambda_i} (-1)^{\lambda_i+1} = \frac{1}{6} * (-6) = -1 \tag{7}$$

The model is local by construction and satisfies the 8 Bell–CHSH inequalities. For instance

$$-2 \leq E(A_1, B_1) - E(A_{-1}, B_1) + E(A_1, B_{-1}) + E(A_{-1}, B_{-1}) = 0 \leq 2. \tag{8}$$

Hence, according to Fine's theorem a joint probability distribution of $A_1, A_{-1}, B_1, B_{-1}$ does exist. The fact that quantum mechanics forbids us to talk about the experiment of measuring $A_1$ and $A_{-1}$, or the experiment of measuring $B_1$ and $B_{-1}$, is irrelevant to the mathematical fact of existence of joint probability distributions reproducing the marginal distributions corresponding to feasible experiments.

In section 5 of [12], Kupczynski rejects the counterexample by rejecting Fine's theorem. Indeed, he recognizes that $E(A_1, B_1), E(A_1, B_{-1}), E(A_{-1}, B_1), E(A_{-1}, B_{-1})$ satisfy the CHSH inequalities but denies the existence of a joint probability, saying

> *We have 4 incompatible experiments, labeled by $(x, y)$, and only 2 outcomes are out-putted in each trial, thus JP [a joint probability distribution] of 4 random variables $A_1, A_{-1}, B_1, B_{-1}$ does not exist.*

He then continues explaining that

> *Lambare and Franco incorrectly conclude: "according to Fine's theorem, a joint probability distribution of $A_1, A_{-1}, B_1, B_{-1}$ exists, although the experiments are incompatible."*

Although Kupczynski's former statements reduce to a denial of Fine's celebrated theorem, it is baffling that he accepts Fine's theorem by explaining at the beginning of the second paragraph of section 5 in [12]

> *Fine demonstrated, that CHSH are necessary and sufficient conditions for the existence of JP of 4 only pair-wise measurable random variables.*

Since, in our counterexample, the CHSH inequalities hold, denying the existence of a JP for the probabilities in such an experiment amounts to rejecting Fine's theorem. Kupczynski's attempt to invalidate the counterexample is contradictory: one cannot reject the counterexample and accept Fine's theorem.

Problems like these arise when we do not distinguish the actual physical situations from mathematical objects appearing in their mathematical descriptions. We can find an analogy in the theory of the classical electromagnetic field. In that theory, the actual physical situation is given by the field magnitudes $\vec{E}$ and $\vec{B}$, while the potentials $\phi(\vec{r}, t)$ and $\vec{A}(\vec{r}, t)$ are part of the formalism but should not be interpreted in a "literal" way. For instance, the electric field is given by

$$\vec{E} = -\nabla \phi - \frac{\partial \vec{A}}{\partial t}. \tag{9}$$

When we use the Coulomb Gauge, we have

$$\phi(\vec{r}, t) = \int_{\mathcal{R}} \frac{\rho(\vec{r}', t)}{|\vec{r} - \vec{r}'|} d^3 r'. \tag{10}$$

If we fix a position $\vec{r}$ very far from the region $\mathcal{R}$ where the source $\rho(\vec{r}', t)$ is distributed, (10) implies that a change in the value of $\rho$ in $\mathcal{R}$ produces an instantaneous effect in the value of the potential $\phi$ at the distant point $\vec{r}$, contradicting the principle of local action. The explanation for this apparent paradox is that, since the scalar potential $\phi$ does not describe an actual physical situation, the principle of locality is not violated.

In our case, the role of the joint probability distribution of $(A_1, A_{-1}, B_1, B_{-1})$ is similar to the scalar potential $\phi$ in the electromagnetic analogy, and it does not possess a direct physical meaning. The actual physical situation is described by some marginals of the joint probability corresponding to the field magnitudes $\vec{E}$ and $\vec{B}$ of the electromagnetic case. Apart from the fact that it contains a second term besides $\phi$, the analog to (9), in terms of probability mass functions, is

$$p_{A_1, B_1}(a_1, b_1) := P(A_1 = a_1, B_1 = b_1) = \sum_{a_{-1}, b_{-1} \in \{1, -1\}} p_{A_1, A_{-1}, B_1, B_{-1}}(a_1, a_{-1}, b_1, b_{-1}), \tag{11}$$

where $a_1 = \pm 1$, $b_1 = \pm 1$. Similar formulae apply for the experimentally meaningful probabilities $p_{A_1, B_{-1}}(a_1, b_{-1})$, $p_{A_1, B_1}(a_{-1}, b_1)$, and $p_{A_{-1}, B_{-1}}(a_{-1}, b_{-1})$. The marginal probability mass functions $p_{A_1, B_{-1}}$ and $p_{B_1, B_{-1}}$ do not have physical meaning and cannot be interpreted literally.

Thus, the existence of a joint probability does not imply the physical measurability of incompatible experiments, just as $\phi$ does not imply action at a distance. Conversely, as proved by the counterexample, the impossibility of simultaneously measuring $(A_1, A_{-1})$ and $(B_1, B_{-1})$ does not hinder the existence of a purely mathematical joint probability.

Of course, a joint probability in a CHSH experiment does not exist when, the experiments violate a Bell inequality because of Bell's theorem. Incompatibility of quantum

observables makes this possible but is not the direct cause. The issues with the joint probabilities are irrelevant to what John Bell taught us, namely, that quantum mechanics—and after loophole-free experiments, also nature—cannot be embedded in a locally causal theory unless we violate statistical independence.

## 3. Kupczynski's Framework

Kupczynki [1,5] incorporates contextual hidden variables, standing for random disturbances arising in the measurement apparatus and dependent on the local measurement setting, as follows. Consider an experiment in which Alice and Bob's settings are $x$ and $y$. To begin with, hidden variables $(\lambda_1, \lambda_2)$ with some arbitrary joint probability mass function $p(\lambda_1, \lambda_2)$, not depending on the local settings $x$ and $y$ chosen by the experimenters, are transmitted from the source to the two measurement stations. At Alice's station and Bob's station, independently of one another, and independently of $(\lambda_1, \lambda_2)$, local hidden variables $\lambda_x$ and $\lambda_y$ are created with probability mass functions $p_x(\lambda_x)$ and $p_y(\lambda_y)$. The measurement outcome on Alice's side is then $A_x(\lambda_1, \lambda_x)$, and similarly on Bob's side, $B_y(\lambda_1, \lambda_y)$. The functions $A_x$ and $B_y$ depend in any way whatever on $x$ and $y$, respectively; even the domains of these functions can vary. The sets of possible outcomes of $\lambda_x$ and $\lambda_y$ may depend on $x$ and $y$, respectively. Now, repeat this story with, instead of $x, y$, settings $x, y'$, then $x', y$, then $x', y'$. In this way, Kupczynski defined the four expectation values $E(A_x B_y)$, $E(A_{x'} B_y)$, $E(A_x B_{y'})$, $E(A_{x'} B_{y'})$ of interest, on four "dedicated" and disjoint hidden variable spaces. Therefore, he says, he is unable to define the "counterfactual" expectation values which are used in a usual proof in the non-contextual case of the Bell–CHSH inequalities. Does this mean that the inequalities need not hold? He states that because of the huge number of free parameters which his model allows, it must be possible to violate the inequalities. He does not actually specify any particular instantiation of all the parameters which does the job.

He claimed that other authors had already done just that. So, how did they do that? Here, a till now down-played feature of Kupczynski's framework comes into play. In [5], Kupczynski's measurement outcomes take three values: $-1$, $+1$, and $0$. This can be mathematically reduced to binary outcome values $\pm 1$ by interpreting his measurement functions $A$, $B$ as representing the conditional expectation values of the outcomes at each measurement station, conditional on the hidden variables carried from the source, and given the settings (a move later taken by Bell as well). His outcome "zero" could be mathematically interpreted as: "no particle was detected; toss an independent fair coin to get a numerical outcome $\pm 1$". As we show in the next section, the CHSH inequality holds for the correlations $E(A_x B_y)$, $E(A_{x'} B_y)$, $E(A_x B_{y'})$, $E(A_{x'} B_{y'})$ produced by the Kupczynski model. After some "redefining" of various mathematical variables one ends up in the standard Bell local hidden variables setup with outcomes $\pm 1$ and $0$, (or as we just remarked, just $\pm 1$) and binary settings $x, x'$ for Alice; $y, y'$ for Bob. The standard simple proof of the CHSH inequality applies.

Thanks to the presence of the further possible outcome $0$, the correlations generated by MK's context-dependent local hidden variables model are not, however, the ones usually of interest to the experimenter. The experimenter, typically trusting quantum mechanics and interested in the quantum mechanical physics revealed in his or her experiment, and not interested in refuting local realism, would estimate the post-selected correlations $E(A_x B_y | A_x \neq 0, B_y \neq 0)$ in the obvious way, by discarding trials with zero outcomes. The problem with this is that, as is very well known, it is possible to "fake" quantum correlations in an entirely local realistic way, provided one tolerates a sufficiently large rejection rate. The experimenter interested in rigorously testing local realism must take account this into account. MK knows this too. He has a section entitled "Subtle relationship of probabilistic models with experimental protocols" in which he repeats what is well known, that (under the model of local hidden variables) the correlations $E(A_x B_y | A_x \neq 0, B_y \neq 0)$ need not satisfy the CHSH inequalities. Curiously, in this section MK also reveals that he knows that Bell inequalities hold for his framework of "probabilistic local causality"; see the Appendix A to this paper.

MK is aware that in recent experiments these loopholes appear to have been avoided. Two of the 2015 "loophole-free" experiments (NIST, Vienna) exploited the fact that technological improvements had given experimenters detection rates above the critical threshold of 66.67%. The other two experiments (Delft, Munich) were actually three-party experiments using the technology of entanglement swapping in order to create "event-ready detectors". In the first case, the crucial theoretical ingredient was provided by the classic results of Eberhard (1993) [15], and, in the second case, by the experimental set-up described by John Bell in his famous paper "Bertlmann's socks and the nature of reality" (Bell, 1981) [3]. The important thing about all four of these loophole-free experiments was that "zero outcomes" did not occur at all, no outcomes were discarded. Admittedly, this is a subtle issue with respect to the Delft and Munich three-party experiments, in which the correlations between Alice and Bob's outcomes are studied conditional on their settings and on a third player, Charlie's, outcome. The spatio-temporal arrangement of the experiment should be such that Charlie's outcome cannot influence Alice's or Bob's settings without action faster than the speed of light.

MK goes on in [5] to discuss various already known weaknesses of the experiments. Indeed, the 2015 experiments were not perfect. Since then, they have been replicated and sharpened; the interest of experimentalists has moved on to possible technological applications of loophole-free experiments (device independent quantum key distribution), where new challenges have to be met. One of the most recent experiments was that of Zhang et al. (2022) [16]. Thanks to improvements in random setting generation, there is no sign of violation of no-signalling in the observed correlations between settings on one side and outcomes on the other. Such disturbing features of the 2015 experiments were likely due to slowly shifting physical properties of the random setting generators. The "hidden variable" time is correlated with what is happening in both wings of the experiment. In the language of applied statisticians, we see *spurious correlations*, due to a *hidden confounder*. How to deal with this problem? Most importantly, improve the random number generation so that there is no longer a shifting bias in setting choices. and, as a surrogate measure (already used in the 2015 experiments), use martingale-based tests and a reasonable assumption that the bias is bounded (on average) by some amount. The experiment [16] also has a really large sample size and, due to being based on entanglement swapping and trapped atoms, there is no detection loophole. There is a locality loophole: Charlie is in the same laboratory as Alice. The Munich team does not yet have enough resources for three laboratory experiments (something which Delft already realized in 2015).

We return briefly to the metaphysical implications of the Bell–CHSH inequalities, and discussion of the physical assumptions behind them, after presenting the central mathematical content of this paper. We show explicitly how one can derive Bell–CHSH inequalities for Kupczynski's model. Contextual setting-dependent parameters do not allow one to escape the Bell–CHSH inequalities.

## 4. The Mathematical Details

Here we continue to use MK's notation from [5]. MK says about his framework: "counterfactual expectations $E(A_x A_{x'})$, $E(B_y B_{y'})$, $E(A_x A_{x'} B_y B_{y'})$ do not exist and Bell and CHSH inequalities may not be derived". He hereby refers to the usual CHSH inequalities for the four expectations $E(A_x B_y)$, $E(A_x B_{y'})$, $E(A_{x'} B_y)$, $E(A_{x'} B_{y'})$. The context is a Bell-type experiment in which Alice chooses between settings $x$ and $x'$, and Bob chooses between settings $y$ and $y'$. MK talks about four different Kolmogorov probability models for the four sub-experiments (one setting choice for Alice and one for Bob). Here are his expressions for the four expectation values of interest, where his already long formulae have been amplified by inserting part of the definition of the four underlying sample spaces $\Lambda_{xy}$, $\Lambda_{xy'}$, $\Lambda_{x'y}$, $\Lambda_{x'y'}$.

$$E(A_x B_y) = \sum_{(\lambda_1, \lambda_2, \lambda_x, \lambda_y) \in \Lambda_{xy}} A_x(\lambda_1, \lambda_x) B_y(\lambda_2, \lambda_y) p_x(\lambda_x) p_y(\lambda_y) p(\lambda_1, \lambda_2),$$

$$E(A_x B_{y'}) = \sum_{(\lambda_1, \lambda_2, \lambda_x, \lambda_{y'}) \in \Lambda_{xy'}} A_x(\lambda_1, \lambda_x) B_{y'}(\lambda_2, \lambda_{y'}) p_x(\lambda_x) p_{y'}(\lambda_{y'}) p(\lambda_1, \lambda_2),$$

$$E(A_{x'} B_y) = \sum_{(\lambda_1, \lambda_2, \lambda_{x'}, \lambda_y) \in \Lambda_{x'y}} A_x(\lambda_1, \lambda_{x'}) B_y(\lambda_2, \lambda_y) p_{x'}(\lambda_{x'}) p_y(\lambda_y) p(\lambda_1, \lambda_2),$$

$$E(A_{x'} B_{y'}) = \sum_{(\lambda_1, \lambda_2, \lambda_{x'}, \lambda_{y'}) \in \Lambda_{x'y'}} A_{x'}(\lambda_1, \lambda_{x'}) B_{y'}(\lambda_2, \lambda_{y'}) p_{x'}(\lambda_{x'}) p_{y'}(\lambda_{y'}) p(\lambda_1, \lambda_2).$$

These four equations are a complicated way to say the following: with settings $x, y$, hidden variables $(\lambda_1, \lambda_2)$ with some arbitrary joint probability mass function $p(\lambda_1, \lambda_2)$ (independent of $x$ and $y$) are transmitted from the source to the two measurement stations. At Alice's station and Bob's station, independently of one another, local hidden variables $\lambda_x$ and $\lambda_y$ are created with probability mass functions $p_x(\lambda_x)$ and $p_y(\lambda_y)$. The measurement outcome on Alice's side is $A_x(\lambda_1, \lambda_x)$, and, similarly, on Bob's side, $B_y(\lambda_2, \lambda_y)$. Now repeat this story with, instead of $x, y$, settings $x, y'$, then $x', y$, then $x', y'$.

Here is just one of many ways to prove that the Bell–CHSH inequalities hold for these correlations. Take as sample space a set of tuples $\boldsymbol{\lambda} = (\lambda_1, \lambda_2, \lambda_x, \lambda_{x'}, \lambda_y, \lambda_{y'})$. This space is just the Cartesian product of the spaces whose existence Kupczynski already hypothesized. Take as probability mass function on this space the product

$$p(\lambda_1, \lambda_2) p_x(\lambda_x) p_{x'}(\lambda_{x'}) p_y(\lambda_y) p_{y'}(\lambda_{y'}).$$

Finally, define new measurement functions $\mathbf{A}(\boldsymbol{\lambda}, x) = A_x(\lambda_1, \lambda_x)$, $\mathbf{B}(\boldsymbol{\lambda}, y) = B_y(\lambda_2, \lambda_y)$ where $x$ can be replaced by $x'$ and/or $y$ by $y'$. Now compute $E(\mathbf{A}_x \mathbf{B}_y)$, also with $x$ replaced by $x'$ and/or $y$ by $y'$. It is immediately clear that the four new expectation values of products have exactly the same values as those just exhibited in Kupczynski's space. We can now go back to Kupczynski's own earlier traditional derivation of Bell–CHSH. There is no barrier to running through the usual proof, since all four expectations of products are defined on the same probability space.

There are more efficient constructions. As one learns in courses on Monte Carlo simulation, one can define a discrete random variable with an arbitrary probability distribution as a function of a single uniformly distributed random variable on the unit interval $[0, 1]$. Thus, one could define $\lambda_x$ and $\lambda'_x$ as functions of a single uniformly distributed random variable $U_1$ and of a second argument $x$ or $x'$. Similarly, define $\lambda_y$ and $\lambda'_y$ as functions of a single, independent, uniformly distributed random variable $U_2$ and of a second argument $y$ or $y'$. We just add to our original $(\lambda_1, \lambda_2)$ two independent random variables $U_1$, $U_2$ and redefine our measurement functions in the obvious way. In this way, we can accommodate any number of setting choices for Alice and Bob without introducing more "contextual" randomness into the two measurement functions. This is an important insight. Contextual randomness does not need randomness dependent on the setting. The setting dependence can be passed into the deterministic part of the model.

Finally a remark. MK mysteriously says "counterfactual expectations $E(A_x A_{x'})$, $E(B_y B_{y'})$, $E(A_x A_{x'} B_y B_{y'})$ do not exist". We do not understand the purpose of this remark. The Bell–CHSH inequalities follow from the existence of a joint probability distribution of four random variables $(A_x, A_{x'}, B_y, B_{y'})$ which contain, as bivariate marginals, the joint probability distributions of $(A_x, B_y)$, $(A_{x'}, B_y)$, $(A_x, B_{y'})$, $(A_{x'}, B_{y'})$. A typical proof of the Bell–CHSH inequalities next considers the random variable $A_x B_y - A_{x'} B_y - A_x B_{y'} - A_{x'} B_{y'}$, which can hereby be constructed. In the case that outcomes take values in $\{-1, +1\}$, one shows that this random variable takes values in $\{-2, +2\}$. In the superficially more general case with outcomes in $[-1, +1]$ one shows that it takes values in $[-2, +2]$. In either case, the expectation value lies in $[-2, +2]$, thus yielding two one-sided Bell–CHSH inequalities for $E(A_x B_y) - E(A_{x'} B_y) - E(A_x B_{y'}) - E(A_{x'} B_{y'})$. Admittedly, it can be confusing to give the same names to some of a number of random variables defined on five different probability spaces.

As is well known, satisfaction of the complete set of eight one-sided Bell–CHSH inequalities, together with the restriction of matching expectations of single variables, and the assumption that all random variables take values in $\{-1, +1\}$, is a necessary and sufficient condition for the existence of the aforementioned coupling. That result is due to Fine [13]. Boole gave a general methodology for deriving results of this kind in his 1854 book *The Laws of Thought*.

## 5. An Alternative Approach

Here, we derive the Bell–CHSH inequalities as an exercise in the modern theory of causality based on Bayes' nets, with causal graphs described by DAGs (directed acyclic graphs). We start by defining a class of models, of which MK's is a special case.

Figure 1 gives us the physical background and motivation for the causal model described in the DAG of Figure 2. How that is arranged (and it can be arranged in different ways) depends on Alice's and Bob's assistant, Charlie, at the intermediate location in Figure 1. There is no need to discuss Bob's role in this short note. Very different arrangements can lead to quite different kinds of experiments, from the point of view of their realization in terms of quantum mechanics.

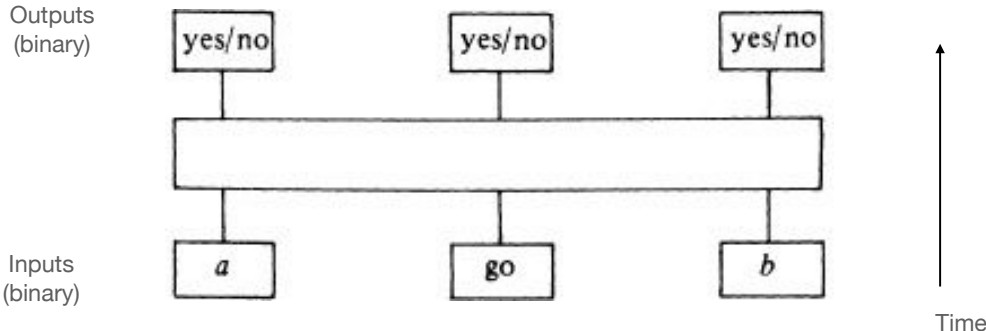

Distance (left to right) is so large that a signal travelling from one side to the other at the speed of light takes longer than the time interval between input and output on each side

One "go = yes" trial has binary inputs and outputs; model as random variables $A, B, X, Y$

**Figure 1.** Spatio-temporal disposition of one trial of a Bell experiment. (Figure 7 from Bell (1981) [3], "Bertlmann's socks and the nature of reality").

Figure 1 is meant to describe the spatio-temporal layout of one *trial* in a long run of such trials of a fairly standard loophole-free Bell experiment. At two distant locations, Alice and Bob each insert a setting into an apparatus, and a short moment later, observe an outcome. Settings and outcomes are all binary. One may imagine two large machines each with a switch on it that can be set to position "up" or "down"; one may imagine that it starts in some neutral position. A short moment later, a light starts flashing (this is the binary *outcome*), which could be red or green. Alice and Bob each write down their settings and their outcomes. This is repeated many times. The whole thing is synchronized (with the help of Charlie at the central location). The trials are numbered, say from 1 to $N$, and occupy short *time-slots* of fixed length. The arrangement is such that, again and again, Alice's outcome is written down before a signal carrying Bob's setting could possibly reach Alice's apparatus, and vice versa.

As explained, each trial has two binary inputs or settings, and two binary outputs or outcomes. We denote them using the language of classical probability theory by random variables $A, B, X, Y$ where $A, B \in \{1, 2\}$ and $X, Y \in \{-1, +1\}$. A complete experiment corresponds to a stack of $N$ copies of this graphical model, ordered by time. We make no assumptions whatsoever (for the time being) about independence or identical distributions. The experiment generates an $N \times 4$ spreadsheet of 4-tuples $(a, b, x, y)$. The settings $A, B$ should be thought of merely as labels (categorical variables). The outcomes $X, Y$

arem thought of as numerical. In fact, we derive inequalities for the four *correlations* $\mathbb{E}(XY|A = a, B = b)$ for one trial.

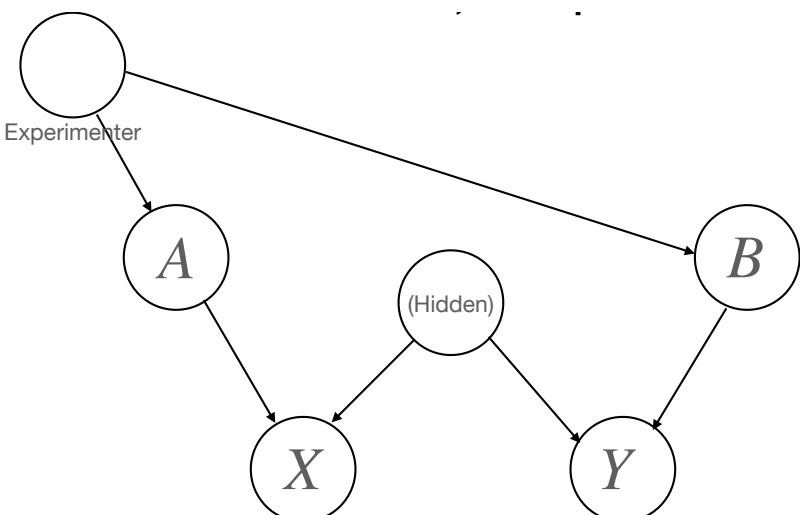

**Figure 2.** Graphical model of one trial of a Bell experiment.

In Figure 2, the nodes labeled *A*, *B*, *X*, *Y* correspond to the four observed binary variables. The other two nodes annotated "Experimenter" and "(Hidden)" correspond to factors leading to the statistical dependence structure of $(A, B, X, Y)$ of two kinds. On the one hand, the experimenter externally has control of the choice of the settings. In some experiments, they are intended to be the results of external, fair coin tosses. Thus, the experimenter might try to achieve a situation wherein *A* and *B* are statistically independent and completely random. The important thing, however, is the aim to have the mechanism leading to selection of two settings statistically independent of the physics of what is going on inside the long horizontal box of Figure 1. This mechanism is unknown and unspecified. In the physics literature, one uses the phrase "hidden variables", and they are thought to be the aspects of the initial state of all the stuff inside the long box which leads in a quasi-deterministic fashion to the actually observed measurement outcomes. The model, therefore, represents a classical physical model, classical in the sense of pre-quantum theory, and one in which experimental settings can be chosen in a statistically independent manner from the parameters of the physical processes, essentially deterministic, which lead to the actually observed measurement outcomes at the two ends of the long box.

Thus, we make the following assumptions. There are two statistically independent random variables (not necessarily real-valued – they may take values in any measure spaces whatsoever), which are denoted by $\Lambda_E$ and $\Lambda_H$, such that the probability distribution of $(A, B, X, Y)$ can be simulated as follows. First of all, draw outcomes $\lambda_E$ and $\lambda_H$, independently, from any two probability distributions over any measure spaces whatsoever. Next, given $\lambda_E$, draw outcomes $a, b$ from any two probability distributions on $\{1, 2\}$, depending on $\lambda_E$. Next, given $a$ and $\lambda_H$, draw $x \in \{-1, +1\}$ from some probability distribution depending on these two parameters, and, similarly, independently draw $\in \{-1, +1\}$ from some probability distribution depending on $b$ and $\lambda_H$.

Thus, $\Lambda_H$ is the hidden variable responsible for possible statistical dependence between *X* and *Y* given *A* and *B*. The other not so hidden variable $\Lambda_E$ stands for the experimental procedure leading to determination of the settings *A* and *B*.

In the theory of graphical models, one knows that such models can be thought of as deterministic models, where the random variable connected to any node in the DAG is a deterministic function of the variables associated with nodes with directed links to

that node, together with some independent random variable associated with that node. In particular, therefore, in obvious notation,

$$X = f(A, \Lambda_H, \Lambda_X),$$

$$Y = g(B, \Lambda_H, \Lambda_Y),$$

where $\Lambda := (\Lambda_H, \Lambda_X, \lambda_Y)$ is statistically independent of $(A, B)$, and $f, g$ are some functions. Moreover $(A, B)$ is statistically independent of $\Lambda$. We can now redefine the functions $f$ and $g$ and rewrite the last two displayed equations as

$$X = f(A, \Lambda),$$

$$Y = g(B, \Lambda),$$

where $f, g$ are some functions and $(A, B)$ is statistically independent of $\Lambda$. This is what Bell called a local hidden variables model. It is absolutely clear that Kupczynski's notion of a probabilistic contextual local causal model is of this form. It is a special case of a non-local contextual model,

$$X = f(A, B, \Lambda),$$

$$Y = g(A, B, \Lambda),$$

in which Alice's outcome can also depend on Bob's setting, and vice versa.

Kupczynski claims that Bell inequalities cannot (or may not?) be derived for his model. However, this is easy. Thanks to the assumption that $(A, B)$ is statistically independent of $\Lambda$, one can define four random variables $(X_1, X_2, Y_1, Y_2)$ as

$$X_a = f(a, \Lambda)$$

$$Y_b = g(b, \Lambda).$$

These four have a joint probability distribution by construction, and take values in $\{-1, +1\}$. By the usual simple proof, all Bell–CHSH inequalities hold for the four correlations $\mathbb{E}(X_a Y_b)$. By statistical independence, however, each of these four correlations is identically equal to the "experimentally accessible" correlation $\mathbb{E}(XY \mid A = a, B = b)$; for all $a, b$

$$\mathbb{E}(X_a Y_b) = \mathbb{E}(XY \mid A = a, B = b)$$

while

$$-2 \ \leq \ \mathbb{E}(X_1 Y_1) - \mathbb{E}(X_1 Y_2) - \mathbb{E}(X_2 Y_1) - \mathbb{E}(X_2 Y_2) \ \leq \ +2$$

and, similarly, for the comparison of each of the other three correlations with the sum of the others.

The whole argument also applies (with a little more work) to the case when the outcomes lie in the set $\{-1, 0, +1\}$, or even in the interval $[-1, +1]$. An easy way to see this is to interpret values in $[-1, +1]$ taken by $X$ and $Y$ not as the actual measurement outcomes, but as their expectation values given relevant settings and hidden variables. One simply needs to add to the already hypothesized hidden variables further independent uniform $[0, 1]$ random variables to realize a random variable with given conditional expectation in $[-1, 1]$ as a function of the auxiliary uniform variable. The function depends on the values of the conditioning variables. Everything stays exactly as local and contextual as it already was.

## 6. What John Bell Already Said About This

As we said earlier, Bell (1975) [2] answered critics of his work, explaining lucidly how many of the criticisms were based on simple misunderstandings. At the end of his paper he wrote

The objection of de la Peña, Cetto, and Brody [17] is based on a misinterpretation of the demonstration of the theorem. In the course of it reference is made to $A(a', \mu), B(b', \mu)$ as well as to $A(a, \mu), B(b, \mu)$. These authors say "Clearly, since $A, A', B, B'$ are all evaluated for the same $\mu$, they must refer to four measurements carried out on the same electron–positron pair. We can suppose, for instance, that $A'$ is obtained after $A$, and $B'$ after $B''$. But by no means. We are not at all concerned with sequences of measurements on a given particle, or of pairs of measurements on a given pair of particles. We are concerned with experiments in which for each pair the 'spin' of each particle is measured once only. The quantities $A(a', \mu), B(b', \mu)$ are just the same functions $A(a, \mu), B(b, \mu)$ with different arguments.

We think it is absolutely clear that what Bell is saying here is that his hypothesis of local hidden variables entails that functions $A(\cdot, \cdot)$ and $B(\cdot, \cdot)$ exist, together with a probability distribution of some variable called $\mu$ with a probability distribution $\rho$, which does not depend on the settings, such that the "observed" joint probability distributions of pairs of variables, which can be observed together, match the probability distributions predicted by the triple $(A, B, \rho)$: two functions and one probability measure. As we have previously shown, there does exist such a triple, which exactly reproduces any of the sets of four probability distributions generated by Kupczynski's model.

Interestingly, neither de la Peña nor his long term collaborator Cetto have changed their minds. In a recent paper [18], de la Peña does not even cite Bell's response to his long-ago paper [17].

A later paper, Bell (1981) [3], makes it absolutely clear what the hidden variable $\mu$ is supposed to stand for, only it is now denoted by $\lambda$.

It is notable that in this argument nothing is said about the locality, or even localizability, of the variable $\lambda$. These variables could well include, for example, quantum mechanical state vectors, which have no particular localization in ordinary space time.

Earlier in the same paper he writes

It seems reasonable to expect that if sufficiently many such causal factors can be identified and held fixed [together, they form $\lambda$] ... $\lambda$ denotes any number of other variables that might be relevant the residual fluctuations will be independent. ... That is to say we suppose that there are variables $\lambda$, which, if only we knew them, would allow decoupling of the fluctuations.

Bell goes on to derive the CHSH inequalities, where now, his functions $A$ and $B$ would be the conditional expectations of the outcome value at each measurement location, conditional on the local setting and the sources of correlation. Thus, Bell's $A(a, \lambda)$ would be Kupczynski's $\sum_{\lambda_x} A_x(\lambda_1, \lambda_x) p_x(\lambda_x)$ with $\lambda \equiv (\lambda_1, \lambda_2)$ and $a \equiv x$.

Very early on, Bell (1971) [19] discussed the issue of placing hidden variables also in the measuring devices. MK actually devotes a large part of a whole section of [5] to this issue, see the Appendix A.

## 7. Conclusions

MK writes: a "... root of quantum non-locality is Bell's insistence that the violation of Bell-type inequalities in SPCE would mean that a locally causal description of these experiments is impossible", then he quotes Bell's famous summary, "in a theory in which parameters are added to quantum mechanics to determine the results of individual measurements, without changing the statistical predictions, there must be a mechanism whereby the setting of one measuring device can influence the reading of another instrument, however remote. Moreover, the signal involved must propagate instantaneously, so that such a theory could not be Lorentz invariant." MK adds "Bell's statement is correct only if one is talking about an ideal EPRB which does not exist," and "imperfect correlations in SPCE may be explained in a locally causal way if instrument parameters are correctly included

in a probabilistic model". However, as we have shown, what MK calls a probabilistic model is mathematically a special case of the classical local hidden variables model of Bell. Bell–CHSH inequalities hold. Violation of statistical independence $p(\lambda \mid a, b) = p(\lambda)$ cannot be obtained as a consequence of including the influence of measuring devices (for another discussion see References [20,21]). It can be true in a situation with non-detection (outcome space $\{-1, 0, +1\}$), conditional on both particles being detected. This is the detection loophole, yet again.

Of course, everything depends on what one exactly means by "locally causal". This is the topic of a wide-ranging discussion in the foundations of physics, and, thus, on the interface between physics and philosophy. There have been numerous attempts to escape Bell's theorem by careful new definitions of words, such as "local", "realism", and "causal". Bell himself did not much use the phrase "local realism". His work concerned the possibility that a theory of *classical* local hidden variables could reproduce (and, therefore, explain in a classical way) the predictions of quantum mechanics, or closely approximate them. He defined in precise mathematical terms what he meant by a (classical) local hidden variables theory. This paper is concerned with correcting common misinterpretations of Bell's work, and, thus, hopefully clearing the air so that real discussions can be better appreciated. For instance, we mention the recent work of Paul Raymond-Robichaud [22], who gives careful definitions of the words "local" and "real", distinguishing the two levels of discourse involved: the world of experimental facts, and the world of a theoretical framework intended to help understand or explain those facts. He goes on to argue that quantum mechanics is both "local" and "realistic", in his terms. Interestingly, he takes the experimental facts which are to be explained by the theory as expectation values, not individual outcomes. Thus, according to him, QM explains the probability distributions of experimentally observable data in a local realistic way, but he says nothing about the locality or realism of the physical realization of individual random outcomes. By restricting the domain of discourse it is more easy to paste labels like "local" and "real" on what is left.

MK attempts to escape Bell's theorem by invoking the detection loophole, but experiments have now closed the detection loophole. Naturally, there is always room to do better experiments still, removing other imperfections, but we feel that it is wishful thinking to suppose this will never happen. Most recently, the experiment of Zhang et al. (2022) [16] seems to be free of the statistical defects of the famous 2015 loophole-free experiments. There is an entirely adequate sample size, improved random setting generation and no problem with spurious violation of no-signaling. There is no locality loophole with respect to Alice and Bob if one assumes that information travels only with the photons generated in the experiment and through their glass fibre cables, though there is if one would allow signaling "as the crow flies" from one laboratory to another. More seriously, Charlie is in the same laboratory as Alice. Some may argue that this experiment could not be repeated (getting similar results) if Alice's and Bob's laboratories could have been located a couple of hundred meters further apart, and Charlie given a third laboratory, thus allowing the glass fibre distance between Alice and Bob (for instance) to be close to the "as the crow flies" distance. One can hope that these short-comings are removed in due time. It seems to us important to do so, because the experimenters are working on the "next level": the use of loophole-free Bell experiments as just one component of Device Independent Quantum Key Description. Thus, the "loophole-free" aspect needs to be perfected. See, for instance, González-Ruiz et al. (2022) [23] on opportunities to improve Zhang et al.'s set up through the use of single-photon "on chip" devices.

Paradoxically, precisely because of Bell's theorem and the experimental violation of Bell–CHSH inequalities in rigorously performed experiments, we also tend to the same metaphysical conclusions as MK: violation of Bell inequalities is strong indication that "entangled photon pairs cannot be described as pairs of socks nor as pairs of fair dice", and "the measurement outcomes are not predetermined".

**Author Contributions:** R.D.G. initiated this project and wrote an initial draft which he shared with J.P.L. It turned out that J.P.L. had already written a critique on another work of Kupczynski together with Rodney Franco [14]. Discussions led to many changes and to the appendix connecting the results to further material in [5]. Both authors stand behind all the content of the entire paper. All authors have read and agreed to the published version of the manuscript.

**Funding:** The authors received no funding for this research.

**Data Availability Statement:** No data was collected or created for the writing of this paper.

**Acknowledgments:** R.D.G. is grateful for numerous discussions with Marian Kupczynski on his work.

**Conflicts of Interest:** The authors declare that the research was conducted in the absence of any commercial or financial relationships that could be construed as a potential conflict of interest.

## Appendix A. Some Logical Issues

In this appendix, we present yet another proof that MK's "contextual approach" confirms the validity of the Bell inequality. Ironically, the proof is already contained in MK's paper in the section, "Subtle relationship of probabilistic models with experimental protocols".

In that section of his paper, MK explains how Bell first derived his inequality averaging over the measuring devices' hidden variables [19]. We reproduce the equations of Bell's approach for convenience

$$
\begin{aligned}
E(A_x B_y) &= \sum_{\lambda_1, \lambda_2, \lambda_x, \lambda_y} A_x(\lambda_1, \lambda_x) B_y(\lambda_2, \lambda_y) p_x((\lambda_x) p_y(\lambda_y) p(\lambda_1, \lambda_2) \\
&= \sum_{\lambda_1, \lambda_2} \sum_{\lambda_x} A_x(\lambda_1, \lambda_x) p_x(\lambda_x) \sum_{\lambda_y} B_y(\lambda_2, \lambda_y) p_y(\lambda_y) p(\lambda_1, \lambda_2)
\end{aligned} \tag{A1}
$$

Setting

$$
\overline{A}_x(\lambda_1) = \sum_{\lambda_x} A_x(\lambda_1, \lambda_x) p_x(\lambda_x) \tag{A2}
$$

$$
\overline{A}_y(\lambda_2) = \sum_{\lambda_y} A_y(\lambda_1, \lambda_y) p_y(\lambda_y) \tag{A3}
$$

We get

$$
E(A_x B_y) = \sum_{\lambda_1, \lambda_2} \overline{A}_x(\lambda_1) \overline{B}_y(\lambda_2) p(\lambda_1, \lambda_2) \tag{A4}
$$

where $|\overline{A}_x| \leq 1$ and $|\overline{A}_y| \leq 1$. Analogous results follow for $E(A_{x'} B_y)$, $E(A_x B_{y'})$, and $E(A_{x'} B_{y'})$, and, therefore, the derivation with hidden variables proceeds as usual.

According to MK, while Bell's Equations (A1), (A2), (A3), and (A4) describe an impossible to implement experiment, his contextual approach corresponds to real experiments. Does this mean that mathematicians need to obtain permission from experimental physicists before proving a mathematical theorem about abstract probability spaces Or that physicists can choose to ignore mathematical truths? MK recognizes that Bell's formulae give the same results as his "contextual model":

*Although the expectations calculated using the Equations (11)–(14) and (19)–(22) have the same values, the two sets of formulae describe different experiments.*

In [5], the above Equations (11)–(14) correspond to MK's contextual model while (19)–(22) describe Bell's model. Kupczynski explicitly states that the calculated expectations of Bell and his contextual model have the same values. The obvious conclusion is that, even if we assume that MK is correct and Bell's formulae are "impossible to implement", they give the same result as his "contextual model", namely, MK's model describing actual experiments, and, hence, these also satisfy the Bell inequality.

The section "Joint Probabilites" in Reference [14] discusses other logical issues regarding Kupczynski's claims on the existence of joint probabilities.

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
