# Peer review of "Kupczynski’s Contextual Locally Causal Probabilistic Models Are Constrained by Bell’s Theorem"

_quantumrep, doi:10.3390/quantum5020032_

Round 1

Reviewer 1 Report

As we read in the mission of   Quantum Reports: Preference will be given to papers with new conceptual advances. It might be a theoretical paper introducing a new paradigm or an experimental paper breaking current technological limits or both a paper reporting an experiment demonstrating new theoretical concepts. The paper of Gill and Lambare (G-L)  does not satisfy these criteria.

Author Response

Please see tyhe attachment

Reviewer 2 Report

The manuscript criticizes a recent work by Marian Kupczynski (MK) which purports that the Bell-CHSH inequality is violated for a contrived hidden variables model he termed “contextually locally causal probabilistic model”. The authors point out various inaccuracies and flawed claims in MK’s works. They show contrary to MK’s belief that his model does imply a joint probability density (pdf) for all of the variables in the Bell-CHSH setting which clearly shows that the underlying Bell inequality cannot be violated. In the last part, they demonstrate a well-known modern construction of MK’s model using directed acyclic graphs.

The manuscript is important for educational purposes for it illustrates how subtleties in the construction of a highly contrived model of hidden variables can lead to misconceptions about Bell’s theorem. I recommend accepting it after a couple of subtle issues are addressed.

Specific comments:

Section 2 starts with an allegedly false statement, that the incompatibility of Alice/Bob pair of observables hinders the formulation of a joint probability for all variables involved, the A,A’,B,B’. I cannot see how this is false in all cases. Namely, I do not see how can one generally formulate an ordinary joint probability density for non-commuting observables. Of course if this cannot be done, no joint for A,A’,B,B’ may exist. Now, it may be that what the authors mean is that in some cases one can *fit* a classical joint probability that will explain the observed outcomes regardless of the incompatibility of A,A’ or B,B’,  assuming the Bell inequality is not violated. To be more specific, for zero-mean observables (<A>=<A’>=0) with eigenvalues +-1, the joint probability is related to

p(A=A’) = P(A=1, A’=1) + P(A=-1,A’=-1) = (1 + <A A’ > )/2

where <A A’> is the quantum mechanical correlator. Now,  if A and A’ are non-commuting then the correlator <A A’> is generally complex-valued. In other words, there may not be a classical joint pdf. If, on the other hand, the correlator is real-valued then perhaps a joint pdf may be fitted.

As far as I can see the state of affairs is this. MK and the authors may not agree simply because the domain of validity of their statements has never been stated. MK’s statement may be correct in general while the authors point to a particular case. There is a another option, that I may have missed the whole point entirely, in which case I would be glad if the authors correct me.

Section 4:

As pointed out by the authors, the construction of the correlators as reproduced from MK’s work do give rise to a joint probability over all the lambdas and so this model cannot in any way violate the Bell-CHSH. MK’s mysterious remark that E[A_x A_{x’}] and E[B_y B_{y’}] do not exist seems likewise flawed.

Reviewer 3 Report

Some shorts like CHSH and JP are not defined in the text. They should be defined the first time they appear in the text, as you did for DAG or MK, using parenthesis or commas.

There is an extra ")" in line 153 in the last expectation value.

In section 2, I believe you should present another counter-example or the same example of the reference you quote because this section is small and you are saying this fallacy is indeed notorious. It is just strange if you write that it is notorious because it was shown or proved in another paper. Or you can change the title of the section.

Another suggestion is to define somewhere in the paper what do you mean by realism and locality. Depending on the literature you can find different definitions for these terms. Does locality mean that information can not propagate faster than light for you and the papers you are criticizing? What about realism?

By the way, I believe you can improve the criticism by changing the way you sometimes refer to the author. I liked the title of your paper, for example. But the title of section 6 might look to some readers that there is a personal dispute when you write that MK did not get right something.  And this thought would be supported by the number of quotations you mention in your work from his papers. 

Round 2

Reviewer 1 Report

                           Referee report and  reply to the authors :

Kupczynski’s Contextually Locally Causal Probabilistic Models are constrained by Bell Theorem

                                by : Richard D. Gill and Justo Pastor Lambare

                                               May 1, 2023

 The authors did not make any significant changes in their manuscript with an exception, that they reproduced and discussed more in detail Lambare-Franco  counter-example (lines 152 -211). In fact, this counter-example is a correct pedagogical example confirming that in Bell (64) local realistic model and in Lambare –Franco counter-example  JP’s exist, but only in the sense  of a probabilistic coupling. This is what we wrote in our first report.  In the revised version of their manuscript and in their reply they try to say the same, but using different words. However, they still do not understand, or they simply don’t want to admit that our statement: Since variables   (A, A′) and (B, B′) cannot be measured jointly, neither N × 4 spread- sheets nor a joint probability distribution of (A, A′, B, B′) existis correct and not fallacious and false. They seem to forget , that there are always two sets of random variables: one set describes finite samples and a scatter of outcomes in particular experiments and another is a part of a probabilistic model (e.g. probabilistic coupling) used to describe random experiment(s) as a whole. We explain it more in detail in attached detailed report. 

Since a JP of (A, A′, B, B′) does not exist, the estimates of pairwise expectations, from 4 incompatible experiments, violate CHSH inequalities 50% of time. Nevertheless, if the outcomes are predetermined as it is assumed in local realistic hidden variable models, then these experiments may be described by appropriate probabilistic couplings and if a size of finite samples tends to infinity the probability of the violation of CHSH inequalities should tend to 0. Since in Bell Tests CHSH inequalities are significantly violated it allows only to reject, with a great confidence, a postulated local realistic probabilistic models/couplings. Contrary to what the authors claim, in their reply, it does not allow to reject Einstein relativistic causality or to evoke quantum magic. 

As the authors easily guessed I was the reviewer 1 of their manuscript, thus it is pointless to pretend the opposite. As a matter of fact, from the beginning I decided to sign my reports in case, if their manuscript was accepted for publication. Therefore, I wrote my extended round 2 report of their revised manuscript in a form of a short note, which I attach to this introduction.

In this note, I explain, that the title, the abstract and several statements in their manuscript are incorrect, misleading and even defamatory. This is why the paper, as it is written, should not be published. Moreover, this paper is neither a research article nor a review and it does not contain any new material, thus, only for this reason it is not suited for Quantum Reports. The authors should take it into consideration, if they plan  to rewrite it and submit it again  to this or to another scientific journal.

Author Response

We have toned down the language in order to remove any trace of emotion or bad feeling or personal animosity. We stand by our mathematical findings and are glad that the paper seems close to being published, so that the wider community of our peers can make up their own minds.

Reviewer 3 Report

The authors have already reviewed all the points that was raised. The paper is suitable for publication.

Author Response

Thank you very much for looking over this yet again.